# Preparation and Structural Analysis of a Water-Soluble Aminated Lignin

**DOI:** 10.3390/polym16091237

**Published:** 2024-04-28

**Authors:** Qi Zheng, Guangzai Nong, Ning Li

**Affiliations:** School of Resources, Environment and Materials, Guangxi University, Nanning 530004, China; zhengqist@163.com

**Keywords:** lignin, ammonia, aminated lignin

## Abstract

Lignin is insoluble in water, thereby limiting its use in the synthesis of adhesives. Therefore, in this study, an aminated lignin compound was prepared through a lignin amination reaction to increase the amount of raw lignin material that can be used in the synthesis of adhesives; moreover, structural analysis was conducted. The main result of this was the introduction of amino groups into phenolic hydroxyl groups in the hydrolyzing lignin from the raw lignin materials, thus generating the product of aminated lignin. The resulting particle sizes were about 100 nm, the average molecular weight was 57,627 g/mol, and the water solubility of the aminated lignin was about 0.45 g/100 mL. Therefore, the water solubility of raw lignin was greatly improved. The proposed reaction mechanism of phenolic hydroxyl groups and carboxylic acid groups in lignin is a reaction with ammonia molecules; thus, the successful introduction of amino groups generated the aminated lignin compounds. Hence, this article enriches the scientific theory of lignin reactions and provides a reference for the widespread application of raw lignin materials in the field of adhesives.

## 1. Introductions

Lignin is a natural high-polymer compound with abundant reserves [1]. It mainly comprises a basic structure of phenylpropane [2] and, as such, can be used as an auxiliary raw material for synthesizing adhesives [3,4]. However, its water-insolubility characteristics limit its widespread use in the synthesis of adhesives; consequently, it accounts for only 5–15% of the total synthetic raw materials in such processes [5,6]. Due to its insolubility in water, lignin exists in the form of solid, small particles [7,8], so it is difficult to achieve a uniform mixing state when it is mixed with liquid reaction materials such as formaldehyde, urea, melamine, and phenols [9]. The synthetic reaction process is affected by this to a certain extent, which influences the structural uniformity and adhesion consistency of the adhesive, thereby leading to cracking and insufficient smoothness of the adhesive surface, reducing the overall quality of the resulting products [10,11]. Consequently, obtaining water-soluble, lignin-modified materials with the ability to uniformly mix with other liquid reaction materials is important for increasing the amount of raw lignin material used in the synthesis of adhesives [12,13].

Introducing hydrophilic groups into raw lignin materials is an effective method for improving the water solubility of lignin-modified materials [14]. Konduri et al. adeptly introduced sulfomethyl groups into lignin. They observed a notable enhancement in its water solubility, subsequently creating avenues for its efficient employment as a cement dispersant [15]. P. Prinsen et al. successfully incorporated site-specific sulfation techniques to augment the water solubility of lignin, ultimately leading to an improvement in its overall processability [16]. The introduction of hydroxyl groups into raw lignin materials not only enhances their water solubility but also augments their chemical reactivity, thereby facilitating their utilization in the preparation of phenolic resin adhesives [17,18,19].

However, the incorporation of sulfur into lignin is believed to hinder the activity of enzymes and microbes during the bioconversion process, which leads to the poisoning of metal catalysts during chemo-catalytic treatment [20,21]. In addition, the water solubility after the introduction of hydroxyl groups remains insufficient. Hence, it is necessary to search for lignin-modified materials with greater water solubility without sulfur elements.

Amino groups (−NH₂) are strongly hydrophilic functional groups [22]; thus, the concept has been proposed of preparing water-soluble aminated lignin compounds through lignin ammonification reactions. Therefore, synthetic experiments and product structural analysis were conducted in this study. The experimental results indicate that the product of the lignin ammonification reaction is aminated lignin, which has good water solubility. Accordingly, this study breaks through the technical barriers in the preparation of lignin-modified materials with good water solubility without introducing sulfur-containing groups, enriches theoretical knowledge in the field of materials science, and provides a new approach for the widespread application of raw lignin materials in the synthesis of adhesives.

## 2. Methods and Materials

### 2.1. Experimental Materials

Industrial alkali lignin (PR) was obtained from Hunan Taigreen Paper Group Hongjiang Paper Co., Ltd. (Huaihua, China); hydrochloric acid (AR) and ammonia (AR) were purchased from Chengdu Kelong Chemical Co., Ltd. (Chengdu, China).

### 2.2. Preparation Method of Fine Lignin

Industrial alkali lignin was dried in an 80 °C blow-dryer for 36 h to obtain absolutely dry alkali lignin. Then, the product was treated with a ball mill for 10 min to obtain small particles of alkali lignin. Next, the small particles of lignin were dissolved in water to obtain an alkaline lignin solution. Subsequently, diluted hydrochloric acid was added for acidification (adjusted to pH = 2), resulting in lignin precipitation. Finally, the mixture was centrifuged for 0.5 h, and the lower sediment was taken and dried to obtain fine lignin particles.

### 2.3. Synthetic Method of Aminated Lignin

Subsequently, 3.0 g of fine lignin particles and 5.0 mL of ammonia water were added to a hydrothermal reactor, in which the temperature was raised to 130 °C. The reaction was held for 1 h to generate an amino lignin solution, which was then dried in a drying oven to obtain the solid aminated lignin.

### 2.4. Product Morphology Analysis

A scanning electron microscope (SEM; Zeiss Sigma 300, Oberkochen, Germany) was utilized to observe the morphology of the product under an accelerated voltage of 3 kV. It was then compared with the morphology of the raw lignin material before the reaction.

### 2.5. Method for Determining the Molecular Weight of the Products

The molecular distribution of synthetic products was determined through gel permeation chromatography (Agilent PL-GPC50, Agilent Technologies, Santa Clara, CA, USA). The test parameters were as follows: the concentration of the test sample was 0.10 mg/mL, the injection volume was 100 microliters, and the distribution coefficient was 14.1000. This describes the parameters of the distribution process or adsorption–desorption process of the components in the fixed and mobile phases.

### 2.6. Methods for Element Analysis of the Product

A matching energy-dispersive spectrometer (EDS) was used to detect and analyze the contents of different elements in the sample, while the morphology of the product was observed using a scanning electron microscope (SEM; Zeiss Sigma 300, Germany) under an accelerated voltage of 10 kV.

For greater accuracy, the contents of the organic elements (C, H, O, N, and S) in the sample were measured through high-temperature combustion and weight adsorption analysis using an organic element analyzer (Elemantar: Vario UNICUB, Langenselbold, Germany).

### 2.7. Methods for Structural Analysis

X-ray photoelectron spectroscopy (XPS; Thermo Scientific K-Alpha, Waltham, MA, USA) was used to test and analyze the composition of the sample. The test parameters were as follows: excitation source: Al Kα ray (hv = 1486.6 eV); beam spot: 400 um; working voltage: 12 kV; filament current: 6 mA; full-spectrum scanning energy: 100 eV; step size: 1 eV; fine-spectrum scanning energy: 50 eV; and step size: 0.1 eV.

### 2.8. Methods for the Thermal Stability Analysis of the Product

Thermogravimetric analysis (TGA, Shimadzu DTG-60(H), Kyoto, Japan) was used to test the thermal stability of the synthetic products and raw lignin materials. About 4–8 mg of the dried sample was put into position using an alumina crucible and heated from 50 °C to 700 °C at a rate of 10 °C/min. The sample was protected by nitrogen gas with a flow rate of 50 mL/min.

### 2.9. Methods for the Water Solubility Analysis of the Product

In a typical solubility experiment, precisely 2.0 g of water was introduced into a 25 mL glass flask. The flask was subsequently submerged in a temperature-controlled water bath (model HH-1, manufactured by Changzhou Yuexin Instrument manufacturing Co., Ltd., Changzhou China), thereby maintaining a temperature fluctuation range of ±1.0 °C. Once the bath temperature attained the target value, approximately 0.01 g of lignin sample was carefully dispensed into the flask. The mixture was continuously stirred under a nitrogen atmosphere and kept at a constant temperature of 30 °C. Additional aliquots of lignin (each equivalent to 0.1 wt% of the solvent) were incrementally added after complete solubilization of the previously added lignin. This process was systematically repeated until no further dissolution of lignin was observed within a two-hour timeframe [23].

## 3. Results and Discussions

### 3.1. Product Morphology

Figure 1 shows a comparison of the morphology of raw lignin materials and ammonia reaction products magnified at 20,000 times. As shown in Figure 1a, the raw lignin material is spherical in shape and partially agglomerated, with a particle size of about 200 nm. In comparison, the aminated lignin obtained through the ammonification reaction appears in smaller particles with particle sizes of less than 100 nm. In addition, the agglomeration is significantly reduced, thus leading to particles that are relatively dispersed, as shown in Figure 1b.

### 3.2. The Molecular Weight of the Generated Product

Based on GPC analysis, the molecular weight distributions of raw lignin materials and aminated lignin products are shown in Figure 2.

As shown in Figure 2, the relative molecular weight of the aminated lignin is concentrated in the small molecule absorption peak of 4000–4600 g/mol, as well as in the large molecule absorption peak of 40,000–250,000 g/mol. Furthermore, the molecular weight of the raw material lignin is also concentrated in the small molecule absorption peak of 4000–4600 g/mol, as well as in the large molecule absorption peak of 40,000–500,000 g/mol. In comparison, no significant change was observed in the content of the small-molecular-weight substances in lignin after the ammonification reaction. However, the content of the large-molecular-weight substances between 40,000 and 100,000 g/mol significantly increased. These test data indicate that the raw material lignin degrades during the ammonification reaction, thus producing lower-molecular-weight aminated lignin.

In addition, calculations using the test system demonstrate that the number average molecular weight of the raw lignin materials is 45,605 g/mol, while the weight average molecular weight is 66,306 g/mol. Upon reflection, the molecular weight distribution is rather wide, as it contains a large amount of macromolecular lignin with a molecular weight of over 300,000 g/mol. Moreover, the number average molecular weight of aminated lignin is 45,358 g/mol, and the weight average molecular weight is 57,627 g/mol.

In comparison, there is an insignificant difference in the average molecular weight between aminated lignin and the original raw lignin materials, yet the average molecular weight is significantly reduced. The fundamental reason for this is that an alkaline catalytic hydrolysis reaction occurs during the ammonification reaction of lignin, thereby causing the hydrolysis of high-molecular-weight lignin, which generates lower-molecular-weight fragments. As a result, there is a decrease in the average molecular weight. However, due to the relatively large number of these small molecular fragments, their contribution to the average molecular weight is unimportant, and, consequently, the variation in the logarithmic mean molecular weight is unimportant [24].

Therefore, the changes in these two types of statistical data on molecular weight indicate that certain high-molecular-weight chains have broken into lower-molecular-weight fragments. These analytical results are consistent with those of the scanning electron microscopy, which are described later on.

### 3.3. Element Compositions

(1)Results of analysis using an energy-dispersive spectrometer (EDS).

Analysis using an energy-dispersive spectrometer (EDS) obtained the content and distribution of C, N, O, and S, as shown in Table 1 and Figure 3. The images show that C and O are uniformly distributed in raw lignin material and aminated lignin, whereas the lower percentage of N content is not clearly distributed in the EDS images. Table 1 shows the decrease in C content after the reaction, which is caused by the mass of the carbon atoms remaining the same while the mass of the product increases by introducing amino groups during the reaction process. More importantly, the nitrogen content significantly increased after the reaction. Indeed, the nitrogen content in lignin is 0.03%, yet the nitrogen content rises to 3.84% in aminated lignin due to the introduced amino groups.

(2)Results of elemental analysis.

An element analyzer was used to examine the elemental composition of the aminated lignin and raw lignin materials to accurately determine the changes in the N element content both before and after the ammonification reaction. The results of the analysis are shown in Table 2. The nitrogen content in the raw lignin materials is very small, being only 0.4%, whereas the nitrogen content in the aminated lignin produced through the ammonification reaction is 4.25%. Therefore, the nitrogen content significantly increases after the reaction. This result indicates that ammonia reacts with lignin to produce aminated lignin.

In addition, the original unsaturation of the lignin molecules is 4.61, while the unsaturation of aminated lignin is 2.96. This change in the data indicates that lignin macromolecules undergo hydrolysis during the reaction process, thus generating smaller molecular fragments and resulting in a decrease in the cyclic unsaturation of the aminated lignin compounds. This analysis further confirms that lignin undergoes hydrolysis during the ammonification reaction.

The formula of lignin can be expressed as C_9_H_10.77_O_4.73_N_0.06_, with the number of carbon atoms being about 150.0 times that of nitrogen atoms. The formula of aminated lignin may be expressed as C_9_H_14.08_O_4.76_N_0.67_, and the number of carbon atoms is about 13.43 times that of nitrogen atoms. The ratio between carbon and nitrogen is reduced from 150 times to 13.43 times, which also identifies the introduction of the amino groups into the raw lignin materials.

### 3.4. Results of the XPS Analysis

An XPS analysis method was used to understand the changes in the contents of various elements and chemical bonds during the reaction process. Figure 4 shows the XPS spectra of the raw lignin materials and aminated lignin, whereby both the raw lignin materials and aminated lignin contain abundant basic elements such as C and O. However, the absorption peak of the intensity of nitrogen atoms in aminated lignin significantly increases. Furthermore, by comparing the ratios of C and N atoms in the raw lignin material to those of aminated lignin, the C/N ratio in the raw lignin material was found to be 30.59, whereas this ratio decreased to 9.19 in aminated lignin. This change indicates the successful introduction of amino groups into lignin, thereby resulting in an increase in nitrogen content.

Figure 5 shows the fine spectra of C1s of the raw lignin material (Figure 5a) and aminated lignin (Figure 5b), as well as the N1s of aminated lignin (Figure 5c). As shown in Figure 5a, the spectra of C1s of the raw lignin materials were fitted with three peaks at 284.8 eV, 286.4 eV, and 288.9 eV, belonging to the C=C/C−, C−O, and C=O peaks, respectively [25,26].

The spectra of C1s of aminated lignin were fitted with four peaks, namely, 284.8 eV, 285.3 eV, 286.6 eV, and 288.9 eV, belonging to the C=C/C −, C−N, C−O, and C=O peaks, respectively, as shown in Figure 5b. The presence of C−N chemical bonds in aminated lignin indicates the successful introduction of amino groups into lignin.

As shown in Figure 5c, the two fitted peaks in the binding energy of the 399.8 eV and 401.8 eV spectra are the C−N bond and −NH_2_/−NH bond, respectively [27]. The existence of these two sets of chemical bonds further proves the successful introduction of amino groups, thereby generating aminated lignin compounds.

### 3.5. Test Results of Thermal Stability

The thermal stability analysis of the raw lignin material and aminated lignin was performed using the thermogravimetric analysis (TGA) technique. As shown in the TGA curves presented in Figure 6, the thermal degradation behaviors of the two materials showed a similar trend in temperature, which was between 50 and 700 °C. However, there are differences between the two curves. Indeed, the raw lignin material experiences significant weight loss in the range of about 140–210 °C, while there is only a slight change in the weight loss of the aminated lignin sample in the same temperature range. This difference shows that the sample of the raw lignin material degrades at a large scale at temperatures between 140 and 210 °C [28]. In contrast, the sample of aminated lignin degrades slowly between the same temperatures, thus clearly indicating that the sample of aminated lignin is more stable than the sample of the raw lignin materials in the same temperature range.

As shown in the DTG curves presented in Figure 6, the first peak of the heat weight loss of the raw lignin material occurs at about 175 °C. As such, a great amount of degradation reactions take place at that temperature. In comparison, the first peak of the heat weight loss of the sample of aminated lignin occurs at about 225 °C. Thus, a great amount of degradation reactions take place at that temperature. The differences between the first two peaks of heat weight loss in the two curves reveal that the thermal properties of the generated aminated lignin product are more stable than those of the raw lignin material, which is consistent with the abovementioned results of the TGA analysis. This important difference in the reaction temperatures that initiate degradation reconfirms the positive effect of introducing amino groups, which enhances the thermal stability of lignin materials. In effect, the addition of amino groups facilitates the formation of hydrogen bonds, which augments intermolecular interactions and subsequently elevates the temperature at which thermal degradation occurs [29].

However, the second peak of the thermal weight loss of the raw lignin material is noteworthy, for it occurs at a temperature of 342 °C, while that of the generated aminated lignin is at a temperature of 332 °C. The two degradation temperatures at this stage are similar due to the breaking of the methyl–aryl ether bond in the molecule [30,31]. The similar temperatures of the second peaks of the thermal weight loss of the two samples demonstrate that the thermal stability of the lignin materials is relatively minor because of the effect of amination in the second stage of thermal degradation.

After reaching a temperature of 400 °C, weight loss is attributed to the subsequent degradation of lignin, whereby the C−C bonds within the lignin are cleaved, thereby resulting in the liberation of small molecules such as H_2_O, CO, and CO_2_ [28]. Finally, at 700 °C, the main component of the residual material is carbon; moreover, the residual carbon rates are 42.12% and 46.58% for the raw lignin material and aminated lignin, respectively. The residual carbon rate for the aminated lignin product is greater than that of the raw lignin materials, further supporting the superiority of aminated lignin in terms of thermal stability [32].

### 3.6. Test Results of Water Solubility

Introducing amino groups into lignin molecules can significantly improve the water solubility of lignin-modified materials. The aminated lignin generated through a lignin ammonification reaction can be dissolved in water; it presents a yellow–brown color, as shown in Figure 7.

According to testing, the water solubility of the aminated lignin compound is about 0.45 g per 100 mL of water. As a result, it effectively improves on the water solubility of the raw lignin materials. This water-soluble aminated lignin can reduce its surface tension in the process of preparing the adhesive and, consequently, allow various reactions for raw materials to uniformly mix. Thus, it effectively improves progress in a polymerization reaction, ensuring the product quality of the adhesive. More importantly, amino groups were introduced into the lignin materials to improve the reaction activity of lignin; therefore, the surface tension is greatly improved in the process of preparing an adhesive.

Hence, this study breaks through the technical barriers in synthesizing lignin-modified materials with good water solubility, additionally introducing amino groups into lignin materials. Accordingly, it provides a technical reference for increasing the amount of lignin-modified raw materials in the field of adhesives.

### 3.7. Discussion of the Mechanisms of the Lignin Ammonification Reaction

As shown in Figure 8, the lignin ammonification reaction is hypothesized to include six steps. This is based on the results of the structural analysis of the product, combined with the theory of alkaline cooking for lignin removal in the pulp- and paper-making process, the theory of alkaline-catalyzed ether bond hydrolysis, the theory of acid–base binding, and the theory of organic amine generation reactions. Research into the mechanism of this chemical reaction enriches the scientific theory of lignin chemical reactions.

(1)In an aqueous solution of concentrated ammonia, β−O−4 ether bonds undergo hydrolysis under alkaline and high-temperature conditions, thereby causing lignin to degrade and generate small lignin fragments [33,34,35]. (The principle of lignin degradation under alkaline catalysis).(2)Subsequently, under alkaline and high-temperature conditions, the methoxyether bonds on the side chains of the lignin molecules undergo hydrolysis, producing methanol and hydroxylated lignin, which contain a large amount of phenolic and fatty hydroxyl groups [36]. (The principle of alkaline-catalyzed ether bond hydrolysis).(3)Subsequently, the phenolic hydroxyl group on the hydroxylated lignin undergoes an acid–base binding reaction with ammonia molecules, thus generating a transition-state intermediate product between the phenolic hydroxyl and ammonia molecules [37,38]. (The principle of acid–base combination).(4)Subsequently, the intermediate product of the phenolic hydroxyl ammonia molecule is unstable and undergoes a dehydration reaction, generating aniline, thereby successfully introducing amino groups and generating aminated lignin [39,40]. (The principles of organic amine generation).(5)In addition, lignin also contains small amounts of carboxylic acids, and these carboxyl groups combine with ammonia molecules to produce carboxylic acid–ammonia transition-state intermediates [41]. (The principle of acid–base combination).(6)A dehydration reaction occurs to produce a structurally stable acid amide group because the transition-state intermediate carboxylic acid–ammonia is unstable and under high-temperature conditions, [42,43]. (The principle of acid–amide-generated reaction).

### 3.8. Discussion of the Innovations and Limitations to This Research

The innovations of this research: A type of aminated lignin was synthesized through the lignin ammonification reaction, which was followed by its structural analysis and a study of the mechanism of its chemical reaction. The aminated lignin product has higher thermal stability and a good water solubility property, which was 0.45 g per 100 mL of water according to testing. Thus, the generated product of the aminated lignin effectively improves the water solubility of the raw lignin materials. The generated water-soluble aminated lignin can be used as a fine material to prepare adhesives. It can reduce the surface tension between it and other raw materials, allowing the uniform mixing of a variety of reaction materials. Therefore, it can improve the progress of the polymerization reaction. At the same time, the aminated lignin product improves the thermal stability of the raw lignin materials, which can also guarantee the quality of the adhesive products. As a result, this research provides a new way to increase the dosage of lignin-modified raw materials in the field of adhesives and enriches the scientific theory of lignin chemical reactions.

However, there are certain limitations to this research: It was discovered that a small portion of the high-molecular-weight aminated lignin did not dissolve completely in water. In contrast, it was suspended on the water’s surface in the water solubility experiment. Therefore, in-depth research on optimizing the preparation process should be carried out to minimize the production of large-molecular-weight aminated lignin and enhance the water solubility of aminated lignin, with the aim of further improving its application performance. In addition, the molecular weight distribution of ammonified lignin dissolved in water has not been investigated; therefore, the relationship between the molecular weight and the water solubility of aminated lignin products is unknown. Nevertheless, the synthetic reaction involves a high temperature and a high pressure, thereby combining with the toxic gas of ammonia. Consequently, it faces the risk of ammonia leakage under high-temperature and high-pressure conditions in the process of a synthetic reaction. To mitigate this potential hazard, the reaction conditions in future experiments should be refined to conduct the ammonification reaction.

## 4. Conclusions

The following conclusion has been drawn through experiments and theoretical analysis: a type of aminated lignin compound was successfully synthesized through the lignin ammonification reaction. The particle size of the aminated lignin is about 100 nm, with an average molecular weight of 57,627 g/mol. Moreover, the agglomeration of the lignin particle size was significantly reduced. Indeed, the aminated lignin has a good water solubility of 0.45 g per 100 mL of water. Moreover, the thermal stability of aminated lignin is better than that of raw lignin materials.

The mechanism of the lignin ammonification reaction is as follows: lignin undergoes a degradation reaction and a demethoxy reaction due to the action of ammonia water, thus generating hydroxylated lignin. Subsequently, the phenolic hydroxyl and carboxylic acid groups on the hydroxylated lignin undergo an acid–base binding reaction with ammonia molecules, thereby generating a transition state of phenol ammonia and carboxylic acid ammonia intermediates. Finally, the phenol ammonia and carboxylic acid ammonia intermediates remove one water molecule each and are transformed into amide groups, successfully introducing amino groups into lignin, as well as generating aminated lignin compounds.

## Figures and Tables

**Figure 1 polymers-16-01237-f001:**
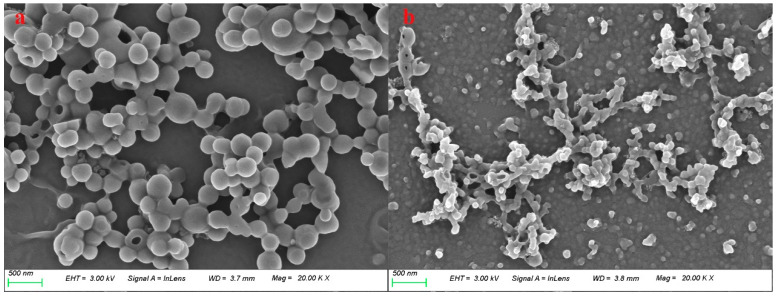
The morphologies of lignin (**a**) and aminated lignin (**b**).

**Figure 2 polymers-16-01237-f002:**
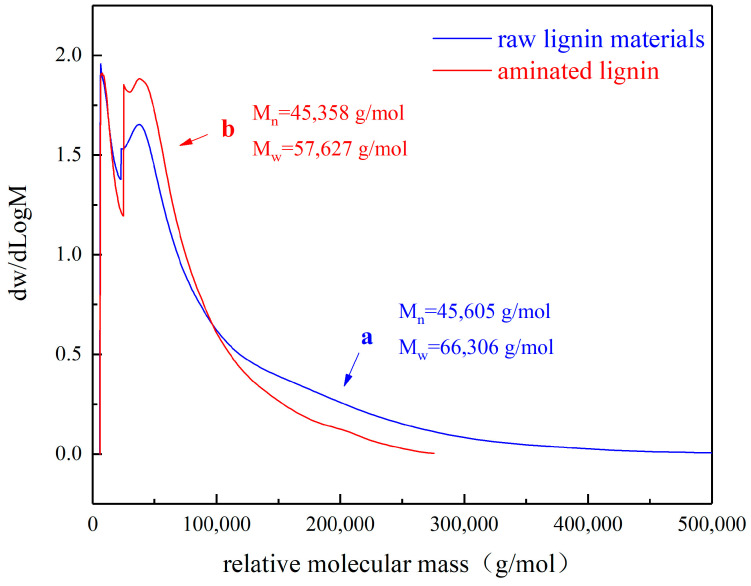
The molecular weight distributions of the generated aminated lignin and the used raw lignin materials.

**Figure 3 polymers-16-01237-f003:**
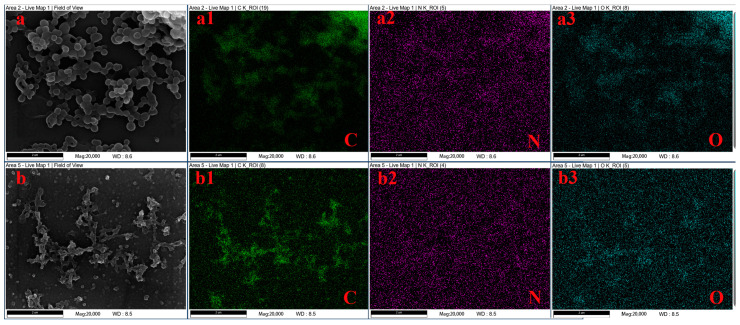
EDS analysis of raw lignin materials (**a**) and aminated lignin (**b**). (**a1**) and (**b1**) are C elements; (**a2**) and (**b2**) are N elements; (**a3**) and (**b3**) are O elements.

**Figure 4 polymers-16-01237-f004:**
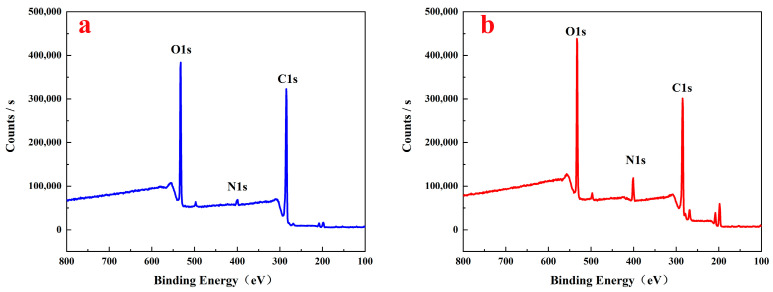
XPS full spectrum of raw lignin material (**a**) and aminated lignin (**b**).

**Figure 5 polymers-16-01237-f005:**
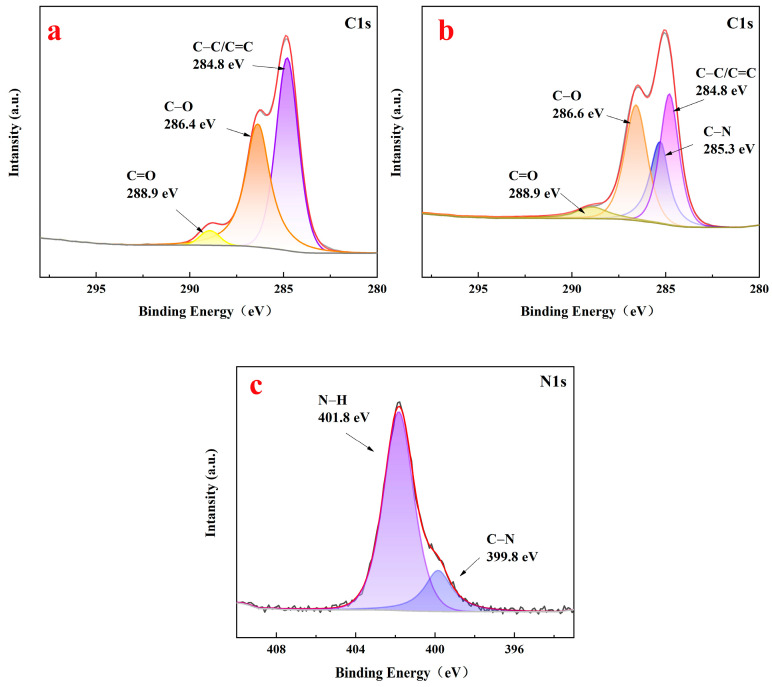
The fine XPS spectra of C1s of raw lignin material (**a**) and aminated lignin (**b**), as well as the fine XPS spectra of N1s of aminated lignin (**c**).

**Figure 6 polymers-16-01237-f006:**
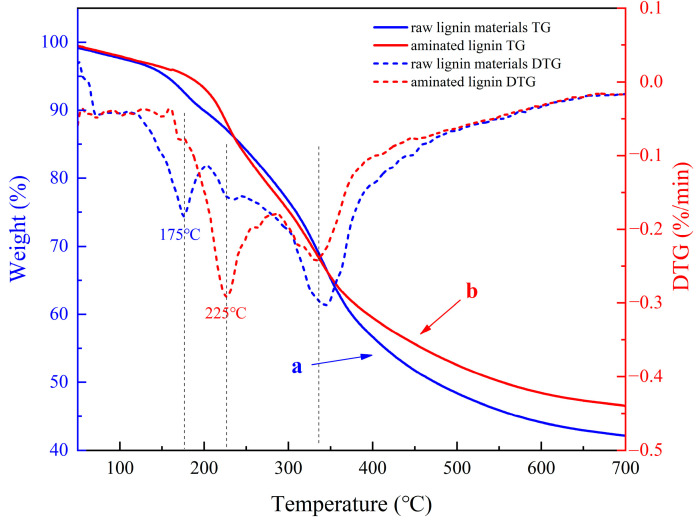
Thermogravimetric analysis (TGA) and derivative thermogravimetry (DTG) of the raw lignin material (a) and aminated lignin (b).

**Figure 7 polymers-16-01237-f007:**

Water solubility of aminated lignin.

**Figure 8 polymers-16-01237-f008:**
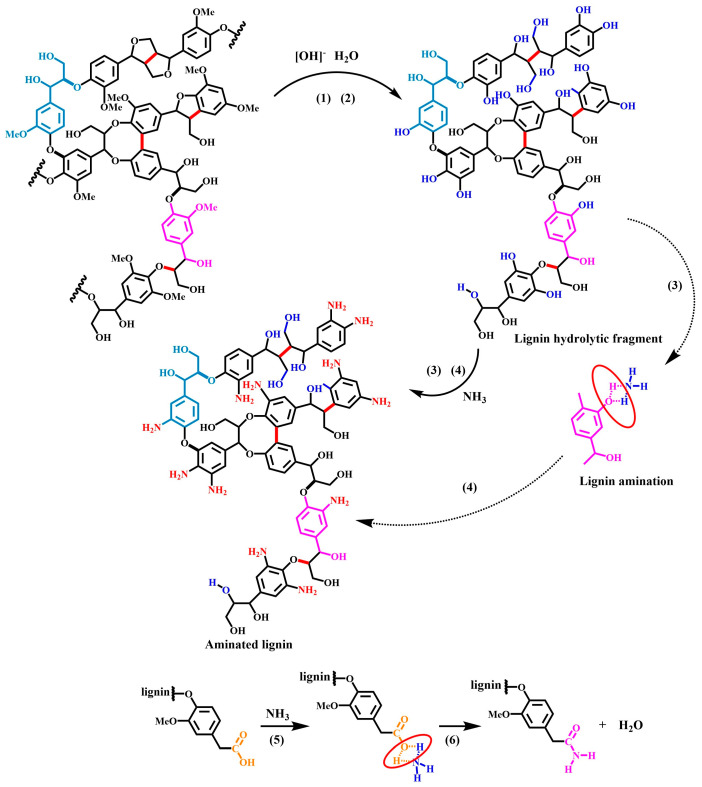
Mechanisms of the lignin ammonification reaction.

**Table 1 polymers-16-01237-t001:** EDS analysis results of raw lignin material and aminated lignin.

Element	Raw Lignin Material		Aminated Lignin	
	Weight %	Atomic %	Weight %	Atomic %
C	84.90	88.23	82.20	85.65
N	0.03	0.02	3.84	3.43
O	15.06	11.75	13.94	10.90
S	0.01	0	0.02	0.01

**Table 2 polymers-16-01237-t002:** Element content of lignin and aminated lignin.

	N( %)	C (%)	H (%)	S (%)	O (%)	Residual	Unsaturation
Lignin	0.40	52.01	5.19	0.14	36.46	5.80	4.61
Aminated lignin	4.25	48.67	6.35	0	34.32	6.41	2.96

## Data Availability

The original contributions presented in the study are included in the article, further inquiries can be directed to the corresponding authors.

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
