# Peer review of "Preparation and Structural Analysis of a Water-Soluble Aminated Lignin"

_polymers, 2024, doi:10.3390/polym16091237_

Round 1
Reviewer 1 Report
Comments and Suggestions for Authors
This manuscript addresses the preparation of a water-soluble aminated lignin to be used as raw material targeting adhesives application. Moreover, this study brings new insights to the lignin aminate materials using the typical reactivity of hydroxyl groups on the natural polymer.
At this stage the manuscript requires some revisions, namely:
Page 1: The abstract section should be considerably improved by providing the main findings. Paragraphs as the indicated in lines 12 and 13 is information relevant for the materials and methods section.
Page 5: In the Element composition discussion of the lignin raw material and after chemical modification, the table 1 and table 2 requires more EDS spectra of each lignin (before and after treatment), to identify statistical relevant differences between elements in the discussion.
Page 7: To support the discussion on the Mechanisms of lignin ammoniation reaction it is also recommended to perform Nuclear magnetic resonance (NMR) techniques, since it constitutes the state-of-the-art for the structural analysis of lignin.
Page 8: In the conclusion section the author states that after the modification the lignin shows a “particle size of the aminated lignin is about 100 nm”, however it should be also referred that in fact, is the agglomeration of the lignin particle size that is significantly reduced. It is recommended to revise the statement.
In general the manuscript was well conduct, however to reinforce this study, this reviewer recommends to perform the previous EDS replicas; 1H NMR, or 13C NMR to complement the chemical characterization and considering the potential application, to access the thermal stability of both lignins by using thermogravimetric analysis (TGA).
Reviewer 2 Report
Comments and Suggestions for Authors
Dear Authors
Lignin is insoluble in water, limiting its use in the synthesis of adhesives. Therefore, in order to increase the amount of lignin raw materials used in the synthesis of adhesives, a kind of aminated lignin compounds was prepared through lignin amination reaction, and conducting structural analysis in this paper. The particle size of the amino lignin is about 100 nm, with the weight average molecular weight of 57,627 g/mol and a water solubility of 0.45g/100ml. The reaction mechanisms were proposed that the phenolic hydroxyl groups of lignin react with ammonia molecules, and thereby successfully introducing amino groups to generate the aminated lignin compounds was claimed by the authors.
The idea of the research is interesting for the readers and has an important impact on the environment and industry.
The following comments may be helpful for the authors to increase the clarity of their research and the reader's benefits.
General comments
1- The lignin structure contains hydroxyl groups which could be modified through a reaction with 2-chloroethyl amine to have amine ethyl lignin.
2- The using of hydrothermal reactor to conduct the ammonification reaction needs high pressure and temperature which raises hazards of leaking ammonia gas.
3- The authors did not perform a full study of the ammonification operational conditions such as ammonia: Lignin ratio, reaction temperature, reaction time, and reaction pressure.
4- The transformation of the lignin carboxylic groups to amine ones is also a chemical route that could be explored.
5- The effect of the animated lignin particle size on the solubility should be investigated and the obtained data could be included.
In conclusion, additional experimental work may be useful to prove the applicability of the authors' idea.
A major revision is recommended.
Comments on the Quality of English Language
A minor revision is recommended.
Round 2
Reviewer 1 Report
Comments and Suggestions for Authors The manuscript entitled “Preparation of a water-soluble aminated lignin and its structure analysis” was considerably improved by the author and presents data that supports and reinforces the findings and the potential of the modified lignin. The authors also answered the previous questions raised by the reviewer. In the opinion of this reviewer, the manuscript can be considered for publication in the present form.
Reviewer 2 Report
Comments and Suggestions for Authors
Dear Authors
Thanks to the authors for their concern in responding to the raised comments satisfactorily.
I can recommend the revised manuscript for publication.
Greetings
Comments on the Quality of English Language
A minor english revision is recommended.